# Frontal Plane Neurokinematic Mechanisms Stabilizing the Knee and the Pelvis during Unilateral Countermovement Jump in Young Trained Males

**DOI:** 10.3390/ijerph20010220

**Published:** 2022-12-23

**Authors:** Kitty Vadász, Mátyás Varga, Balázs Sebesi, Tibor Hortobágyi, Zsolt Murlasits, Tamás Atlasz, Ádám Fésüs, Márk Váczi

**Affiliations:** 1Institute of Sport Sciences and Physical Education, Faculty of Sciences, University of Pécs, 7624 Pécs, Hungary; 2Somogy County Kaposi Mór Teaching Hospital, 7400 Kaposvár, Hungary; 3Department of Kinesiology, Hungarian University of Sports Science, 1123 Budapest, Hungary; 4Center for Human Movement Sciences, University Medical Center Groningen, University of Groningen, 9700 RB Groningen, The Netherlands

**Keywords:** vertical jump, pelvic tilt, hip abduction, knee valgus, gluteus medius, trained males

## Abstract

(1) The unilateral countermovement jump is commonly used to examine frontal plane kinetics during unilateral loading and to identify athletes with an increased risk of lower limb injuries. In the present study, we examined the biomechanical mechanisms of knee and pelvis stabilization during unilateral vertical jumps. (2) Healthy males performed jumps on a force plate with the dominant leg. Activity of the dominant-side gluteus medius and the contralateral-side quadratus lumborum and erector spinae muscles was recorded with surface EMG. The EMG data were normalized to the EMG activity recorded during maximal voluntary isometric hip abduction and lateral trunk flexion contractions. During jumps, the propulsive impulse was measured, and the pelvis and thigh segmental orientation angles in the frontal plane were recorded and synchronized with the EMG data. (3) The magnitude of knee valgus during the jump did not correlate with hip abduction force, but negatively correlated with gluteus medius activity. This correlation became stronger when gluteus medius activity was normalized to hip abduction force. Propulsive impulse did not correlate with any neuromechanical measurement. (4) We conclude that hip abduction force itself does not regulate the magnitude of knee valgus during unilateral jumps; rather, the gluteus medius should be highly activated to increase frontal-plane knee joint stability.

## 1. Introduction

Athletes frequently perform one- and two-legged vertical jumps as a performance test and as part of plyometric exercise training. The biomechanical characterization of vertical jumps includes knee and hip joint loads that can exceed 3.5 and 5.5 times body weight when both feet are in contact with the ground [1]. In contrast, joint loads in triple jumps reached 15.2 times the body weight during the step phase [2]. In addition, knee joint reaction and vertical ground reaction forces reached 8.5 N·kg^−1^ and 6.2 N·kg^−1^ during single-leg jumps in plyometric training and volleyball take-offs [3,4].

The unilateral countermovement jump (UL-CMJ) is a commonly used test to examine frontal plane’s biomechanics during unilateral loading and to identify athletes with increased risk of leg injuries [5]. The high angular acceleration-generated knee and hip joint torques make the lower extremity joint and the pelvis prone to injury. When transferring from a two-legged stance to a one-legged position, the shift of the center of gravity in the frontal plane produces an asymmetric and imbalanced body position [6]. Due to the lack of bilateral support of the pelvis, torque is generated in the hip joint of the standing leg, which causes a tilt in the pelvis [7]. To compensate, the lumbar spine joints rotate around the sagittal axis. Because of the aforementioned segmental displacements in UL-CMJs, pelvis and lumbar spine stabilizer muscles are important for alleviating excessive frontal-plane joint angular changes, which are linked to an increased risk of injury [8]. Dysfunction of the gluteus medius (GM) can contribute to pelvic tilt even during low-intensity walking [9]. Erector spinae (ES) activation also increases to stabilize the spine by limiting axial rotation. The quadratus lumborum (QL) is an extensor and lateral flexor of the trunk [7] and lifts the iliac crest towards the thoracic cage as weight shifts to the contralateral foot. QL also helps stabilize the pelvis along with the GM when weight is transferred from one leg to the other [10]. Therefore, it is expected that QL and GM activation contributes to pelvic tilt during UL-CMJ, and that pelvic tilt would in turn affect lumbar ES activation.

Although the GM is primarily a hip abductor, an indirect role for stabilizing the knee in the frontal-plane has been also suggested; previous research has found that the magnitude of GM activity plays an important role in determining knee and pelvic stability in the frontal plane [11]. Suboptimal activation of the GM may induce excessive knee valgus during a single-leg squat. Research results have suggested that, among other muscles, the hip abductor muscle’s function may play a significant role in controlling frontal-plane movements of the knee. During single-leg squats, the hip abductor muscles may provide stabilization of the femur, therefore reducing the medial collapse of the knee. This is an important component in the prevention of knee injuries. Still, there is no strong evidence that maximal voluntary hip abduction force and/or GM activation regulate knee valgus during UL-CMJ [12].

The aim of the present study was to examine muscle activations and hip and knee joint kinematics during UL-CMJ. We hypothesized that (i) all three measured muscle activations as well as hip abduction maximal voluntary force are related to the magnitude of pelvic tilt and that (ii) hip abduction maximal voluntary force and GM activation contribute to the magnitude of knee valgus during UL-CMJ. We also determined the contribution of these variables to joint stability while performing jump. We addressed these questions using quantitative dynamometry, electromyography (EMG), and segmental motion tracking while healthy young adults performed UL-CMJs. Identifying the contribution of the aforementioned muscles to the development of unhealthy postures such as pelvic tilt and knee valgus is important, and would provide practitioners and physiotherapists with applicable information in injury prevention.

## 2. Materials and Methods

### 2.1. Participants

Healthy male physical education students (n = 25, age: 21.2 ± 1.3 years; height: 182.4 ± 7.3 cm, training history: 9.7 ± 2.6 years) were recruited from the university to participate in the study. Participants pursued ground-contact sports ~2.9 sessions/wk at the club level. Inclusion criteria were male gender and a minimum of six months of experience in plyometric training. Exclusion criteria were current injuries and past surgeries in the spine, hip, knee, and ankle joints, or pain of orthopedic origin. Participants received a verbal and written description of the experimental protocol, were informed of the potential risks, and provided written informed consent according to the Declaration of Helsinki. The University Ethics Committee approved the protocol (approval number: 7961-PTE2019).

### 2.2. Experimental Protocol

Participants reported to the biomechanics laboratory one time for a 1-hour-long session. They warmed up by riding a cycle ergometer at a self-selected speed for 5 min and performed a few minutes of whole-body and specific stretching of the lower extremity muscles. Participants then performed lateral trunk flexion and hip abduction maximal voluntary isometric contractions (MVC) and completed the testing session by performing UL-CMJs on a force plate using the dominant leg, determined by self-reporting [13].

### 2.3. MVC Testing

MVCs were performed to determine maximum surface EMG activity. Lateral trunk flexion MVC was measured in a lateral lying position with the jumping leg in contact with the floor. Two investigators manually stabilized the hip and the shoulders to remain on the floor. The electric activity of QL and ES was measured on the side opposite the jump leg (Figure 1a).

For maximal voluntary hip abduction force, participants lay on their side but with the jump leg superior. Participants were instructed to perform isometric hip abduction at 0° joint angle with maximum effort. A hand-held dynamometer (C.I.T. Technics, Haren, The Netherlands) was placed 5 cm above the external malleolus to measure hip abduction force (Figure 1b). GM electric activity was measured on the jump leg’s side.

For both lateral flexion and hip abduction MVCs, participants performed two submaximal warm-up trials followed by two maximal effort trials with a one-minute rest between trials. Hip abduction force was normalized for body weight. Verbal encouragement was given to ensure participants’ maximum effort.

### 2.4. UL-CMJ Testing

The UL-CMJs were performed on a force plate (Tenzi, Pilisvörösvár, Hungary) (Figure 2). The participants stood on one leg, with arms akimbo and the knee of the free leg slightly flexed. Participants performed three UL-CMJs with one minute of rest between trials. Participants were instructed to jump as high as possible with hands kept on the hips but received no instructions concerning jumping strategy. Because participants had extensive experience in UL-CMJs as part of plyometric training, the posture of the free leg had little variation between participants. Still, we standardized the posture of the free leg during UL-CMJ and allowed participants to practice the jumps. In this process, we corrected any deviation from the standard position of the free leg as was the case during the measurements. During jumps, vertical ground reaction force was measured (sampling frequency: 420 Hz) and from the force–time curve the propulsive impulse was calculated as follows:$$\vec{J}=\int_{t_{1}}^{t_{2}}\vec{F}dt$$
where $\vec{J}$ = propulsive impulse, $\vec{F}$ = force acting on a body over a time interval *t*_1_ to *t*_2_. Values were then normalized to body mass. The jump trial with the highest propulsive impulse was included in the statistical analyses.

### 2.5. Surface EMG

EMG data were collected telemetrically during all MVC and UL-CMJ tasks. The skin was carefully prepared by shaving and cleaning with alcohol. Dual Ag/AgCl surface electrodes (Noraxon, Scottsdale, AZ, USA) were positioned on the GM of the jump leg, on the QL, and the lumbar ES on the side opposite the jumping leg. EMG electrodes were positioned on ES and GM muscles according to SENIAM recommendations (www.seniam.org, accessed on 3 May 2021). For QL, we followed the methodology of Monteiro et al. [14]. EMG signals were collected (Noraxon, Scottsdale, AZ, USA, sampling frequency: 2000 Hz) and the raw data were processed with the root mean square technique, using a 50-ms moving window. EMG data were filtered using a 20 to 500 Hz bandpass filter.

### 2.6. Kinematic Analyses

A 3D motion capture system (Noraxon, Scottdale, AZ, USA, sampling frequency: 100 Hz) was used for measuring hip and knee joint kinematics during jumps. Combined accelerometer–gyroscope–magnetometer sensors were placed on the shank, thigh, and pelvis, using Velcro straps. We followed the manufacturer’s recommendations in sensor positioning and calibration (www.noraxon.com, accessed on 28 May 2021). The sensors provided frontal plane orientation angle data concerning time during UL-CMJ. With this procedure, we were able to quantify the magnitude of pelvic tilt and knee valgus. Knee angle data were obtained from the orientation angles of two sensors (placed on the shank and the thigh) [15].

### 2.7. Data and Statistical Analyses

All kinematic and EMG signals received form the wireless sensors were synchronized with a versatile synchronization system (MyoSync, Noraxon, Scottsdale, AZ, USA), which is used in case of multi-device setup. For data processing, we used myoRESEARCH 3.18 software (Noraxon, Scottsdale, AZ, USA). During UL-CMJs, peak orientation angles (pelvic tilt and knee valgus) and peak EMG data were collected, considering the entire eccentric and concentric phase of the UL-CMJ, using the knee joint position data as a reference (Figure 3). All jumps’ EMG data were normalized to EMG activity measured during either hip abduction or trunk lateral flexion MVCs. The ratio of hip abduction force to relative GM activation obtained during UL-CMJ was also determined.

Data are reported as mean ± SD. According to the Shapiro–Wilk test, knee valgus, pelvic tilt, and hip abduction force variables were normally distributed. UL-CMJ propulsive impulse, relative GM EMG activity/hip abduction force activation ratio, as well as all three measured muscle activities were not normally distributed, therefore these values were log-transformed. Relative EMG activities of the muscles during UL-CMJ were compared with a one-way analysis of variances (ANOVA), with Bonferroni correction for post-hoc comparison. Pearson product–moment correlations were used to determine associations among all biomechanical and EMG variables. The statistical significance was set at *p* ≤ 0.05. In addition, reliability analysis was performed using intraclass correlation coefficients (ICC) across three UL-CMJ repetitions and the two MVC trials to confirm that measurements were stable among participants. An ICC below 0.50 indicated poor, 0.50–0.75 indicated moderate, 0.75–0.90 indicated good, and above 0.90 indicated excellent reliability [16]. SPSS, Version 25.0 (SPSS Inc, Chicago, IL, USA) was used for all statistical analysis.

## 3. Results

Table 1 shows the reliability data. The mean knee valgus was 6.06° (±8.21), pelvic tilt was 6.94° (±3.06), UL-CMJ propulsive impulse was 2.71 (±0.028) N·s·kg^−1^, and hip abduction force was 3.96 (±0.80) N·kg^−1^.

The EMG activity during the jump was normalized to the EMG activity measured during the MVC test contraction, and the ES muscle showed the highest activity, which was 421%. QL activity was 301% and GM activity was 140% of the values recorded during lateral flexion and hip abduction MVCs, respectively. The one-way ANOVA revealed that the relative EMG activity of the three muscles during UL-CMJ differed from one another (*p* = 0.001, F = 7.53, statistical power = 0.93). The post-hoc analysis revealed that ES activity was significantly greater than GM activity (*p* < 0.0001). QL activity did not differ from any of the other two muscles (Figure 4).

The magnitude of knee valgus did not correlate with hip abduction force during UL-CMJ but negatively correlated with GM activity (*p* ˂ 0.05) (Figure 5 and Table 2).

This correlation became stronger when GM activity was normalized to hip abduction force (GM activity/hip abduction force ratio) (*p* ˂ 0.05) (Figure 6 and Table 2).

Table 2 summarizes the correlation coefficients among all measured and calculated neurokinematic variables. We found that the magnitude of pelvic tilt did not correlate with hip abduction force or with any muscle activity during UL-CMJ. UL-CMJ propulsive impulse correlated with both ES and QL activity (*p* ˂ 0.05) but did not correlate with either GM activity or with the magnitudes of pelvic tilt and knee valgus.

## 4. Discussion

The main finding of the present study was that participants with greater knee valgus exhibited less GM activity and that the magnitude of pelvic tilt was independent of either ES or QL activation during UL-CMJ. Furthermore, our data show that the hip abduction force itself did not regulate the magnitude of knee valgus during UL-CMJ. Finally, UL-CMJ propulsive impulse correlated neither with knee valgus nor with the magnitude of pelvic tilt.

Hip abduction maximal force did not correlate with the magnitude of knee valgus measured during UL-CMJ. This result contradicts a previous report suggesting that larger GM is related to lower knee valgus angle during a drop vertical jump [17]. The discrepancy between the current and this previous study might be related to fact that Ueno et al. [17] compared females to male subjects who performed bilateral jumps as opposed to unilateral jumps in our study. Another study also reported strong negative correlations between the magnitude of knee valgus angle and hip abduction force during single-leg landing tasks among physically active female participants whereas in males the relationship was weaker [18]. Similarly, a negative correlation between knee valgus and hip abduction peak torque (r = −0.37) was also reported during single-leg squat [12]. In our study, the hip abduction maximal force probably was not significantly correlated with knee valgus because our participants were physically active male athletes and the knee valgus was small. It seems that hip adduction force and valgus become correlated in females in particular.

Despite of a lack of significant correlation between hip abduction force and knee valgus during UL-CMJ, in our study, GM activity measured during UL-CMJ and GM activity/hip abduction force ratio each negatively correlated with the magnitude of knee valgus. These associations suggest that probably not abduction force, but the activation of the hip abductor muscle (GM) is the variable that plays the main role in knee valgus in our young, trained male cohort. The importance of GM activation during unilateral lower-extremity tasks is supported by previous data showing that single-leg drop landing from a 30 cm-high box compared with double-leg landing increased knee valgus angle from 1.5 to 8.2° along with a threefold increase in the activity of the GM compared to double-leg drop landing in healthy males [19]. Therefore, it seems that in single-leg jumps, the GM has to be strongly activated and this activation can reduce knee valgus, potentially protecting the knee from injury. Normative values of knee valgus angle in a physically active population during unilateral step landing suggest that knee valgus angle should be symmetrical in the two knees within the range of 5–12° for females and 1–9° for males [20]. In our study, we obtained an average of 6° knee valgus angle (range 2–17°). Excessive knee valgus is related to knee injuries [21]. All in all, our kinematic and EMG data show that isolated hip abduction force itself is not an insightful outcome relative to knee valgus control. However, hip abduction activation does seem to be an important control of knee valgus position in the context of UL-CMJ. For example, participants with 333–351% normalized GM activity produced only 2–4° valgus. In contrast, knee valgus as high as 16–17° corresponded to only 98–139% of normalized GM activity.

We found no relationship between ipsilateral GM and contralateral (non-jumping leg) QL and ES muscle activity and the magnitude of pelvic tilt, perhaps suggesting a more complex muscle activation mechanism being responsible for frontal plane pelvic stability. Little is known about the role of QL in athletic movements. QL counteracts forces generated by abdominal muscles and it stabilizes and mobilizes the spine and pelvis. It has less of a role in lumbar stabilization [22]. ES stabilizes the spine in the sagittal plane and thus is strongly active during the UL-CMJ compared to lateral flexion. GM acts from the femur to stabilize the pelvis and keeps the trunk erect when standing on one leg, running, and walking when one foot is off the ground [23]. A previous study demonstrated that the size of the QL muscle was significantly related to lower limb injury in male elite football players [24]. Willson et al. [8] reported a relationship between hip and trunk muscle activity and lower extremity movement. Accordingly, decreased trunk stability can increase susceptibility to lower extremity injury, and core stability and strength training for the trunk can reduce the risk of a knee injury. Previous research shows 35.0° ± 5.1° anterior pelvic tilt in women during a single-leg triple hop test [25]. Another study found 44.6° ± 10.1° anterior pelvic tilt in male athletes after ACL reconstruction during the propulsive phases of the single-leg vertical jump [26]. GM and QL control pelvic tilt during unilateral movements [27]. Once, however, the pelvis becomes forcibly tilted, frontal-plane rotations in lumbar vertebrae are evoked, possibly increasing the stabilization role of ES. ES functions mainly as a trunk extensor; however, the lateral parts stabilize the spine in the frontal plane. The present data show very low pelvic tilt (7°) in our participants. Since we found no correlation between the activation of ES, QL, and pelvic frontal plane tilt, we conclude that other muscles such as multifidus and transverse abdominis may contribute to frontal plane pelvic tilt.

Propulsive impulse did not correlate with knee valgus, pelvic tilt, and hip abduction force during the UL-CMJ performed on the force plate. In a previous study, we showed that participants with higher hip abduction force produced greater UL-CMJ propulsive impulses. Lanza et al. [28] also confirmed the importance of hip abductor maximal voluntary force and GM rapid activation in the two-legged hop test and the four-square step test for improved performance. These data suggest that perhaps proper femur or/and pelvis positioning during UL-CMJ contributes to better jump performance. In the present study, we found no support for this theory as smaller knee valgus did not produce higher UL-CMJ propulsive impulse. In our previous study [29], we were unable to measure segmental kinematics. Thus, we had no knee valgus data during UL-CMJ. In our present investigation, the mean knee valgus was only 6°. Had we tested participants with high valgus values, UL-CMJ performance could have been substantially greater.

All three investigated muscles showed very high normalized activity in the present study, but ES showed the highest activity of the three during UL-CMJ. Some participants exceeded a sevenfold value when normalized to MVC during lateral trunk flexion, and, at the same time nearly fivefold in QL. We used the lateral flexion test to examine the mechanism of how frontal plane pelvic tilt is controlled during UL-CMJ. However, it seems ES and QL were barely activated during the lateral flexion test. Even though ES and QL were active during lateral flexion, Andersson et al. [30] did not correlate this activation with pelvic frontal plane tilt. Their data imply that spine and pelvic stabilization in other planes might still be significant.

While we failed to observe that ES and QL activation contributed to pelvic tilt, the positive correlation between ES and QL activity and UL-CMJ propulsive impulse assigns an impart role to these muscles in UL-CMJ performance. To confirm, a previous study reported a positive association between ES strength and jump height [31]. Mills et al. [32] also found an increase in maximum vertical jump height after 10 weeks of lumbar–pelvic stability training.

One limitation of the present study was that we examined only males even though it is known that knee valgus values tend to be much greater in females [33,34]. Another important limitation was that we enrolled healthy individuals whose knee valgus is limited compared with those with a history of knee injury. Indeed, individuals with a previous knee injury (with anterior cruciate ligament reconstruction) had a higher knee valgus during single-leg hopping than healthy controls [35]. Finally, we used no interventions (i.e., fatigue, exercise training) to manipulate the variables examined and the nature of the relationship between the muscle activation abduction force and valgus angle.

## 5. Conclusions

In young, moderately trained males, GM activation instead of peak hip abduction force seems to contribute to knee valgus during UL-CMJ. Thus, individuals with high hip abduction force could still produce large knee valgus. We conclude that in addition to adequate force production, an optimal level of GM activation is needed for knee joint stability. We propose that practitioners increase GM activation through functional tasks besides using isolated hip abduction resistance exercises to reduce dynamic knee valgus and knee injury risks.

## Figures and Tables

**Figure 1 ijerph-20-00220-f001:**
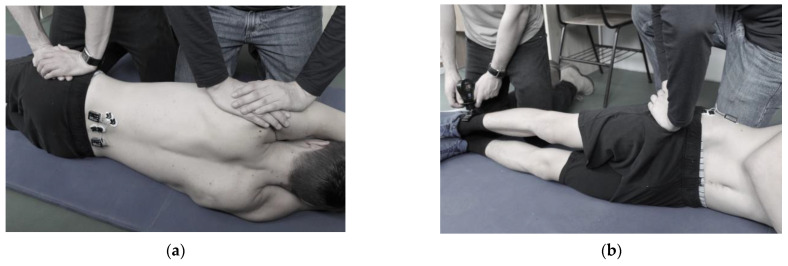
Testing for determining reference EMG values: (**a**) lateral flexion; (**b**) hip abduction.

**Figure 2 ijerph-20-00220-f002:**
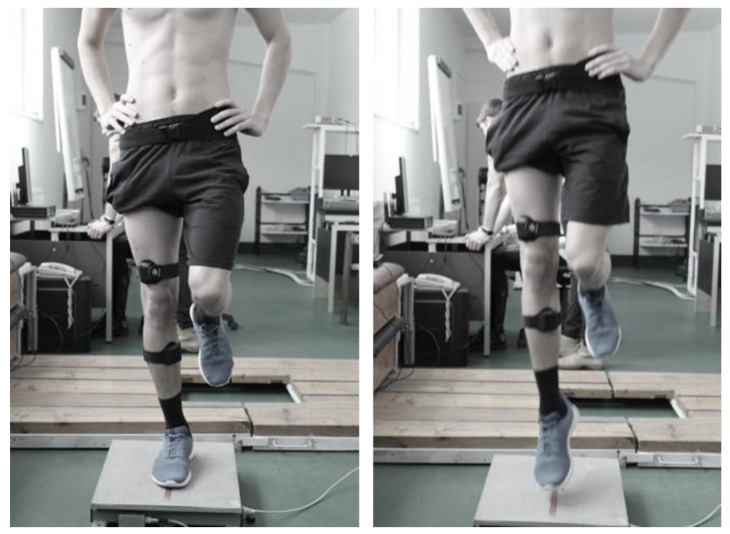
Neurokinematic and ground-reaction force measurements in UL-CMJ.

**Figure 3 ijerph-20-00220-f003:**
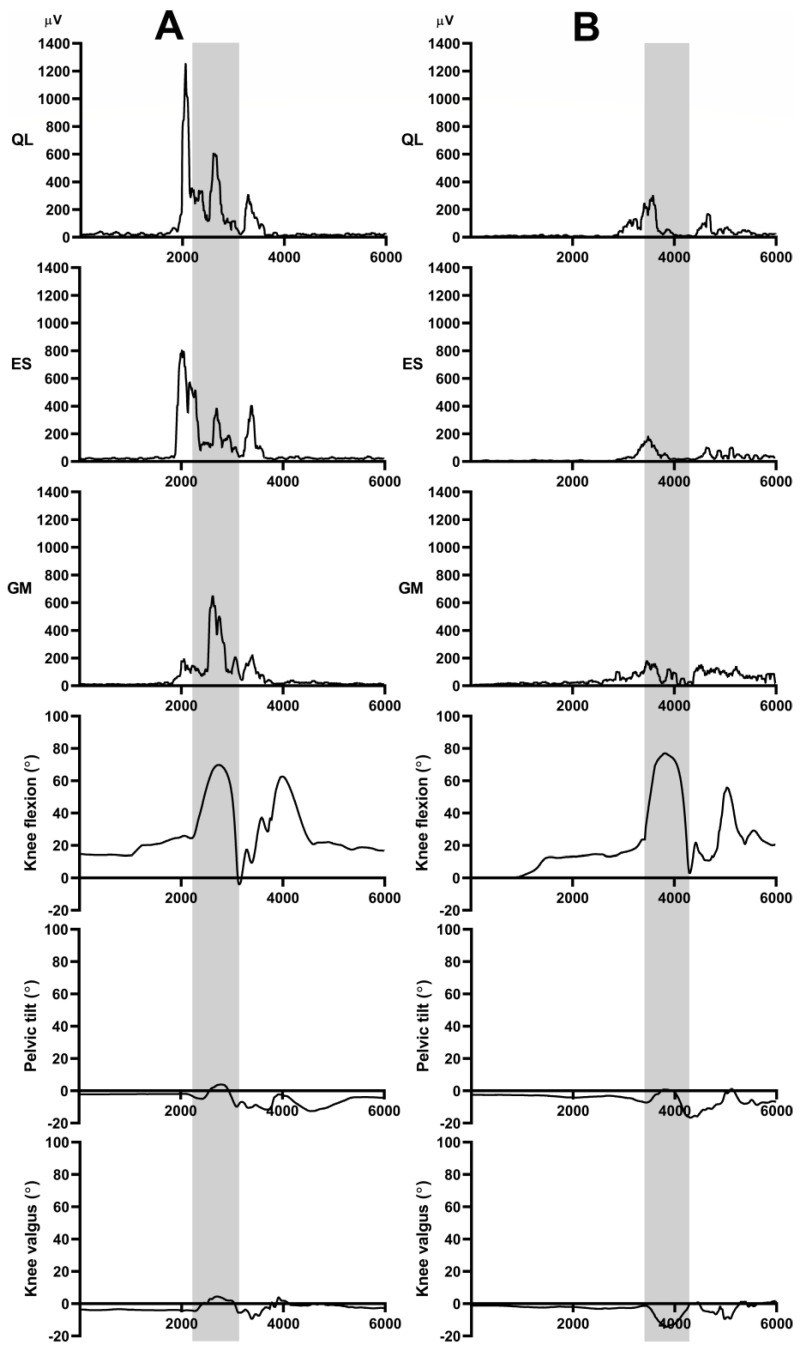
Representative data of the processed EMG signals (RMS) during UL-CMJ. On the *Y* axis, the joint segmental angular change (°) and the recorded muscle EMG activity (µV); on the *X* axis, the time (ms) was identified in each figure. Measurement periods for quadratus lumborum (QL), erector spinae (ES), gluteus medius (GM) muscles during UL-CMJ for a participant with high GM activity and small knee valgus (**A**), and participant with low GM activity and big knee valgus (**B**). Grey area indicates time period of UL-CMJ, involving both eccentric and concentric phases, determined from the knee angle data.

**Figure 4 ijerph-20-00220-f004:**
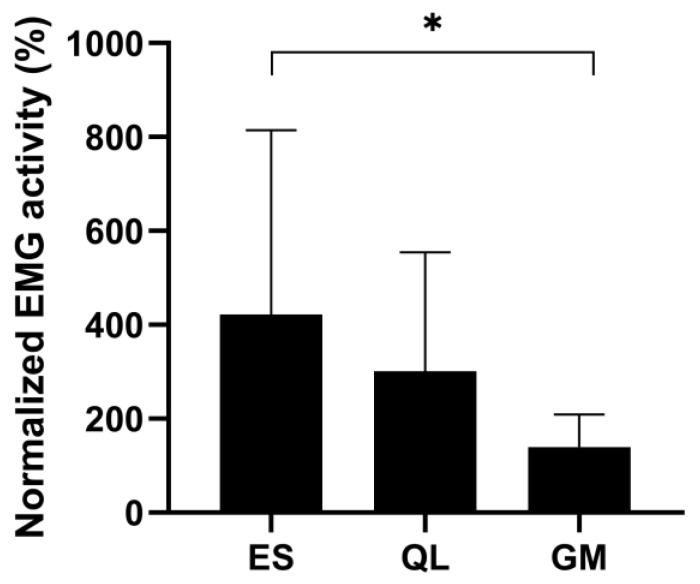
Normalized EMG activities were obtained from GM, QL, and ES muscles during UL-CMJ. * Significant difference (*p* ˂ 0.05).

**Figure 5 ijerph-20-00220-f005:**
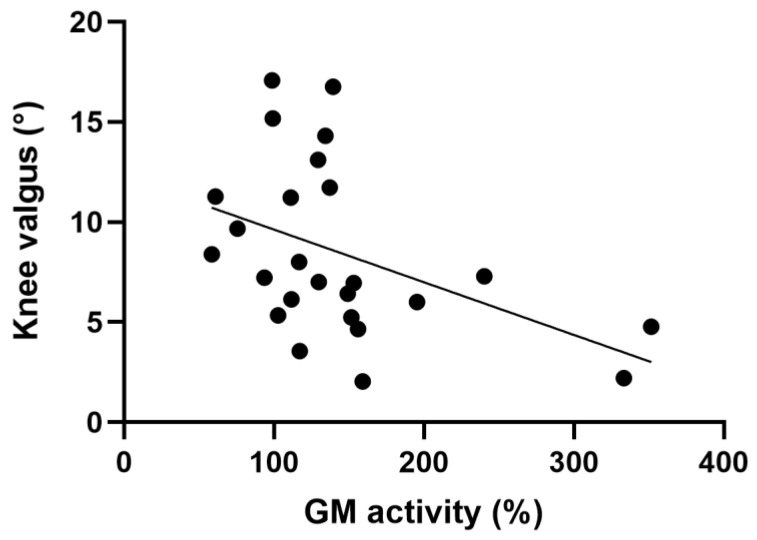
Correlation between the magnitude of knee valgus and GM activity during UL-CMJ (n = 25).

**Figure 6 ijerph-20-00220-f006:**
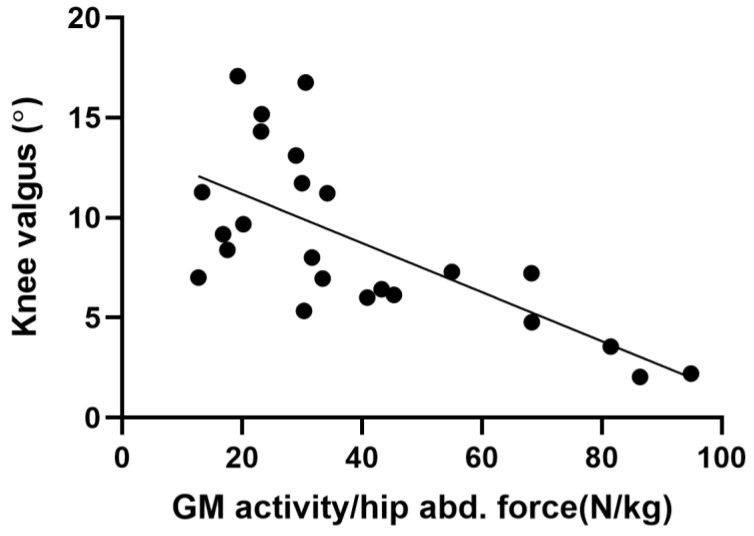
Correlation between the magnitude of knee valgus and GM activity/hip abduction force ratio during UL-CMJ. (n = 25).

**Table 1 ijerph-20-00220-t001:** Reliability data for the measured EMG and kinematical variables (n = 25).

Measured Variables	UL-CMJ ICC	MVC ICC
EMG ES	0.460	0.965
EGM QL	0.816	0.735
EMG GM	0.608	0.836
Knee valgus	0.976	
Pelvic tilt	0.869	
Propulsive impulse	0.986	

UL-CMJ = unilateral countermovement jump; ICC = intraclass correlation coefficient; MVC = maximal voluntary isometric contraction; EMG ES = EMG data of m. erector spinae; EMG QL = EMG data of m. quadratus lumborum; EMG GM = EMG data of m. gluteus medius.

**Table 2 ijerph-20-00220-t002:** Pearson product–moment and Spearman rank correlations among the neurokinematic and kinetic data (n = 25).

	I	KV	PT	F_abd_	GM/F_abd_	GM	ES
KV	0.01						
PT	0.10	0.31					
F_abd_	0.16	0.46	−0.33				
GM/F_abd_	−0.07	−0.56 *	−0.23	−0.17			
GM	0.13	−0.45 *	−0.10	0.10	0.85		
ES	0.43 *	−0.25	−0.13	−0.04	0.36	0.45 *	
QL	0.42 *	0.00	−0.13	0.12	−0.14	0.11	0.46 *

I = UL-CMJ propulsive impulse, KV = knee valgus, PT = pelvic tilt, F_abd_ = hip abduction force, GM/F_abd_ = relative gluteus medius EMG activity/hip abduction force ratio, GM = gluteus medius, ES = erector spinae, QL = quadratus lumborum, * significant at *p* < 0.05.

## Data Availability

Not applicable.

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
