# Peer review of "Frontal Plane Neurokinematic Mechanisms Stabilizing the Knee and the Pelvis during Unilateral Countermovement Jump in Young Trained Males"

_ijerph, 2022, doi:10.3390/ijerph20010220_

Round 1

Reviewer 1 Report

The authors study the stabilization during unilateral vertical jumps. The authors use EMG to measure muscle activation and a force plate to measure the pulse. The knee valgus was also measured by the sensor attached to the body. The authors perform statistical analyses on the experimental results and found the level of gluteus medius activation contributes more to the knee valgus.  The paper provides enough background and uses the right tools in the experiments. The analyses look reasonable to me.  Several comments should be addressed before publishing.

1.       So many abbreviations. The authors should provide a table about abbreviations for readers.

2.       Looking at Fig.2, I wonder if the position of the leg that does not jump can influence the results.

3.       How do you sync the data from different sensors in Fig.3

4.       Fig.3, provides an X-axis label and makes the Range reasonable.

Reviewer 2 Report

Comments:

-          Abstract: it does not like a standard scientific paper. Number (1)-(4) should not be numbered. It should include background, purpose, material, method, results, etc.

-          Introduction: Literature review [1-12] did not carefully analyzed. Each study should be discussed deeply. From that, the authors can provide the new motivation/contributions of the present study.

-          The results of this study are only focused on the ANOVA statistical test. The findings are relatively poor.

-          The participants are collected from the university for measuring but how to validate/verify the achieved findings. It’s still unclear.

last comments, I note that:

- Method to conduct this study is not well presented. It's flaw. - Samples on participanrs are not enough. - Results are still poor. - new contribution is unclear in introduction.

Round 2

Reviewer 1 Report

The authors have addressed my concerns

Reviewer 2 Report

The paper is more understanding for readers. It can be accepted in the present form. No comments.